# A Nonlinear Radio-Photon Conversion Device

Irina L. Vinogradova [1], Azat R. Gizatulin [1,*], Ivan K. Meshkov [1], Anton V. Bourdine [2,3,4] and Manish Tiwari [5]

1   Department of Telecommunication Systems, Ufa State Aviation Technical University, K. Marxa Street, 450000 Ufa, Russia; vil-4@mail.ru (I.L.V.); mik.ivan@bk.ru (I.K.M.)
2   Department of Communication Lines, Povolzhskiy State University of Telecommunications and Informatics, 23 Lev Tolstoy Street, 443010 Samara, Russia; bourdine@yandex.ru
3   JSC "Scientific Production Association State Optical Institute Named after Vavilov S.I.", 36/1 Babushkin Street, 192171 Saint Petersburg, Russia
4   Department of Photonics and Communication Links, Saint Petersburg State University of Telecommunications named after M.A. Bonch-Bruevich, 22 Bolshevikov Avenue, 193232 Saint Petersburg, Russia
5   Department of Electronics and Communication Engineering, School of Electrical and Electronics and Communication Engineering, Manipal University Jaipur, Ajmer Road, Jaipur 303007, India; manish.tiwari@jaipur.manipal.edu
*   Correspondence: azat_poincare@mail.ru

**Abstract:** The article analyzes existing materials and structures with quadratic-nonlinear optical properties that can be used to generate a difference frequency in the terahertz and sub-terahertz frequency ranges. The principle of constructing a nonlinear optical-radio converter, based on an optical focon (a focusing cone), is proposed. Based on the assumption that this focon can be implemented from the metal-organic framework (MOF), we propose a technique for modeling its parameters. The mathematical model of the process of propagation and nonlinear interaction of waves inside the focon is based on a simplification of the nonlinear wave equation. Within the framework of the developed model, the following parameters are approximately determined: the 3D gradient of the linear refractive index and the function determining the geometric profile of the focon, which provide a few-mode-based generation of the difference frequency. The achieved theoretical efficiency of radio frequency generation is at least 1%; the proposed device provides a guiding structure for both optical and radio signals in contrast to the known solutions.

**Keywords:** radio photonics; radio-over-fiber; orbital angular momentum; quadratic-nonlinear structure; difference frequency generation

## 1. Introduction

One of the main strategic tasks for the development and implementation of new generation communication networks is the development of new information technologies that allow increasing the bit-rate of data transmission (up to 1 Tbit/s) and the spectral efficiency (60 bit/s/Hz) of the used radio frequency channels, as well as the use of ultra-wideband communication channels (up to 100 GHz), high energy efficiency with ultra-short network propagation delays up to 1 μs. These modern-day system requirements can be addressed by improved data processing techniques based on new principles of access to the physical layer of the network, modulation schemes and channel coding, waveforms, reception of the information, multiple access and security in communication channels. Thus, according to one of the pioneer books dedicated to 6G system analysis and vision [1], which proposes possible ways to implement broadband access in 6G wireless networks, it is assumed that sub-terahertz radio frequencies will be useful for wireless communication and scanning in the THz range. In these applications, it is required to ensure the operation of non-extended radio channels, but with a significant information capacity not only between subscribers, a subscriber and a device, but also between smart devices.

These channel requirements imply both the use of new modulation formats and physical layer technologies, which ensure, in particular, the "seamlessness" (absence of

intermediate operations) of optical-radio conversion, which is especially relevant in sub-THz and THz systems, where it is very difficult to generate high-frequency radio signals by electronic devices, and it also seems inefficient to decode a multichannel signal for subsequent conversion of its components [2–5]. Therefore, to ensure such seamlessness, in particular, for optical-radio conversion in two-frequency (or wavelengths λ: $\lambda_1$, $\lambda_2$) radio over fiber (RoF) networks transmitting vortex signals (carrying orbital angular momentum, OAM), a nonlinear converter was developed [2]. However, according to the results of the simulation of the converter's parameters [2] and experimental results (see Section 2), the efficacy of this converter is very low—about $10^{-16}$—which obviously is not applicable (note that we do not use any resonators such as whispering gallery modes (WGM) resonator because of complexity of fiber-to-resonator coupling, although it could increase the converter efficiency). According to our study, this is due to (a) a very small coefficient of quadratic nonlinearity $\chi^{(2)}$ of the ferroelectric crystal used in [2], and it is also (b) due to radiation losses in the specified crystal because of divergence of the input light beam, which is associated with the absence of any wave-guiding structures in the scheme proposed in [2]. The negative impact of these factors can be significantly reduced if the material and structure for the nonlinear element are properly selected. Thus, the purpose of this article is to determine the structure and the material for this type of frequency conversion.

Note that a number of papers have been published describing the generation of high-frequency radio signal by mixing optical waves with close frequencies (so-called photomixers). They generate difference frequency corresponding to the radio range (subTHz, THz) due to the interaction of input optical waves in a nonlinear element. However, according to publications, in particular [6–8], a THz photomixer consists of two independent tunable laser sources that provide a difference frequency at the output of the device by heterodyning beams on a photoconductive receiver, where charge transfer occurs in a semiconductor material (GaAs is used in the [6–8]). That means the usage of electronic circuits, which significantly reduces the positive effect achieved, given the frequency of the discussed radio range, while the proposed solution does not require any additional electronic equipment. It should be noted that the basic scheme of the photomixer presented in [6–8] corresponds to the classical RoF scheme, which uses two independent lasers and which is known to have a significant phase noise—there is a need of phase-noise compensation. Moreover, in contrast to photomixers, the proposed device is supposed to guide both optical and radio signals, see Section 3.

## 2. Analysis of Optical Quadratic-Nonlinear Materials and Structures Potentially Suitable for the Design of a Nonlinear Converter

It was shown in [2] that during the nonlinear interaction of two input light beams in a quadratic medium, for example, in a lithium niobate (LiNbO$_3$) crystal, an electromagnetic beam with a frequency equal to a difference between the frequencies of the input signals is generated (difference frequency generation, DFG). In this case, several physical properties of the input beams are also transferred to the output beam (conserved), namely, for OAM carrying input beams with topological charges $\ell_1$ and $\ell_2$, respectively, at the output we obtain OAM signal with $\ell_3$, which is equal to the difference between the input OAMs: $\ell_3 = |\ell_1 - \ell_2|$ [2]. This fact arouses interest in further development of the seamless principle of optical-radio conversion by a nonlinear element, presented in [2].

However, numerical estimates in [2] showed that the generation efficiency η of a vortex (OAM) radio beam for a nonlinear medium with a length of several millimeters is extremely low (about $10^{-16}$). This efficiency significantly depends on the frequency ratio between the generated (difference) frequency $\omega_3$ and input frequency $\omega_1$, i.e.,

$$\eta \approx \left(\frac{\omega_3}{\omega_1}\right)^2,$$

(1)

where $\omega_3 = |\omega_1 - \omega_2|$. Some causes of this were emphasized in the introduction: a small value of $\chi^{(2)}$ and the absence of a guiding structure.

Analysis of the other authors results obtained in a similar problem shows that this scheme is actively studied and successfully tested experimentally. Thus, in [9], the conversion of an OAM signal from the optical to the terahertz spectral region was experimentally studied using a quadratic-nonlinear element. The relations for the output signal parameters, in particular, the OAM topological charge ($\ell_{THz}$) and the conversion efficiency $\eta^{DFG}$, are presented as Equations (2) and (3).

$$\ell_{THz} = (\ell_{OPA1} - \ell_{OPA2}) \frac{\lambda_{OPA2} - \lambda_{OPA1}}{|\lambda_{OPA2} - \lambda_{OPA1}|}, \tag{2}$$

$$\eta_{0,l}^{DFG}(\omega_1, \omega_2) = \left| \iint u_{0,l}^{LG}(r, \phi, \omega_{DFG})^* u_{DFG}(r, \phi, \omega_1, \omega_2) r dr d\varphi \right|^2, \tag{3}$$

where *OPA* corresponds to the optical signal, $u$ is the field function, $\lambda$ is wavelength, and $\omega$ is frequency. In this case, the conversion efficiency, according to [9], significantly depends on the beam interaction region, but the dependence on the input and output frequencies ratio, see Equation (1), is not taken into account. The authors of [10,11] also assume that a quadratic ferroelectric crystal is quite applicable for the process of parametric frequency generation: terahertz difference frequency, half-frequency, as well as double frequency and sum-frequency. In this case, the efficiency of the output beam depends on the parameters of the system and is proportional to the propagation length $z$:

$$I_2 = \frac{\omega^4 \pi^2}{4} \beta_2^2 I_1^2 z^2 \sin c^2 \left[ (k_2 - 2k_1) \frac{z}{2} \right]. \tag{4}$$

The coefficients in Equation (4) are described in [11]. The authors of [12] also present the results of experimental approbation of the considered principle of frequency transformation. According to the studies carried out, it was found in [12] that a quadratic-nonlinear crystal can indeed be used for the considered task; however, the conversion efficiency is also low—it did not exceed 0.1% of the input radiation intensity.

Having analyzed the published works in the area under consideration [9–12], our scientific team has also performed an experimental study of this principle of generation using a periodically poled lithium niobate crystal—PPLN [4,5]—and based on the model presented in [2] (see Figure 1). In this setup, we used two lasers with output powers of about 22 dBm, whose signals (at close continuous wavelengths 1550 nm and 1550.8 nm, respectively) are fed separately to two branches of free-space setup (Figure 1b); one of the free-space branches has a diffractive optical element (DOE), forming an OAM signal (namely, a branch with signal at 1550 nm); both optical signals are coaxially (by free-space beam splitter) focused through a lens on the heated PPLN (T~200 °C) and the radio signal from its output is detected in the radio domain by a waveguide antenna for the mm range (Figure 1b). It was possible to establish that the quadratic-nonlinear element used actually provides the generation of a difference frequency even when the phase matching of the waves is performed only approximately—by custom PPLN structure with poling period about 800 μm that corresponds to the desired output frequency—100 GHz; PPLN has size 10 × 2 × 2 mm (Figure 1a) and is doped by MgO to improve efficacy. However, the intensity of the output radio signal remains at the noise level (about −55 dBm), which significantly limits the practical usability of the considered principle. We should note that in all analyzed papers on this topic, the authors make the same conclusion: the efficiency of the difference frequency generation using this type of crystal is unacceptably low. Our theoretical approach is described in detail in [2]; in any case, the results of our experiment are not the subject of this article.

At the same time, the physical reasons for this low efficiency, generally speaking, have not been studied in detail, but only assumptions have been put forward, which, moreover, differ from work to work. The general assumptions of the authors who published the results of independent studies are as follows: firstly, existing ferroelectric crystals have a small value of $\chi^{(2)}$, secondly, optical beams arriving at the input end of the crystal have a diver-

gence, inverse proportional to the radiation wavelength. This leads to significant decrease in the beam intensity—the diffraction divergence causes a decrease in the optical pump power flux, which reduces the efficiency of difference frequency generation. Moreover, one can list additional factors such as (1) the absence of a resonant propagation condition for the generated radio frequency; (2) violation of the few-mode propagation regime for the input optical radiation inside the crystal (this is in addition to the divergence). If the high-order modes are excited, the mode purity is violated and therefore efficacy decreases.

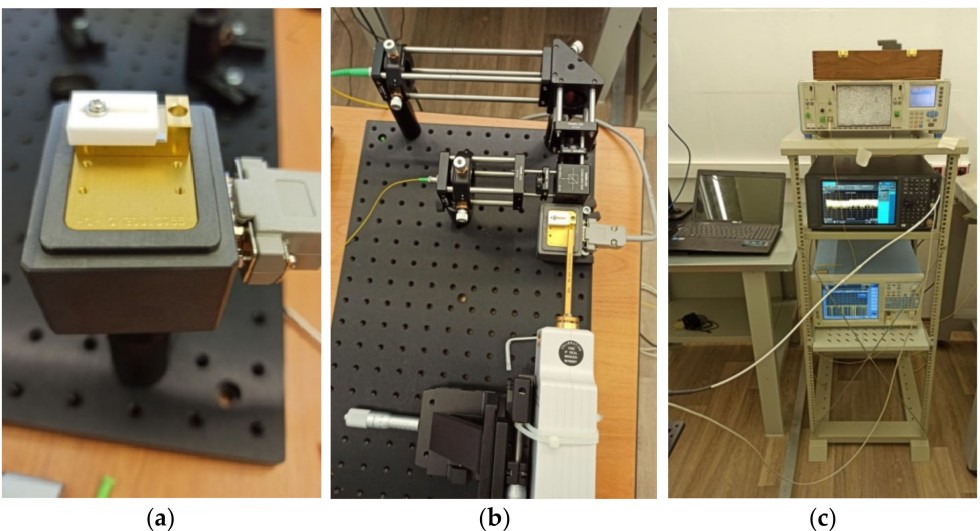

| (a) | (b) | (c) |

**Figure 1.** Photos of the fastening system (**a**–**c**) of a ferroelectric lithium niobate crystal on an experimental setup using a specialized heating interface and free-space setup (**b**), designed to increase the efficiency of the crystal (the heating driver is not shown; heater (**a**) is open); (**c**) measurement setup including both optical and radio spectrum analyzers and a light source (laser).

Indeed, ferroelectric crystals, such as $LiNbO_3$, $LiTaO_3$, $KNbO_3$, $KTaO_3$, $PbTiO_3$, and the like, have such advantages as [13]: (1) low dependence of light transmission on frequency in the visible and near-IR radiation; (2) low cost and high availability. However, at the same time, they have the following disadvantages: (1) the value of the nonlinear susceptibility $\chi^{(2)}$, defining the efficiency of nonlinear processes, is in the range $\cong (0.5 \ldots 45)'10^{-12}$ m/V, i.e., it is very small in comparison with other quadratic-nonlinear materials that exist today; (2) these materials cannot be made in the form of a long waveguides similar to optical fiber, which will complicate the interface of the converter with fiber optic communication lines.

The small value of $\chi^{(2)}$ in ferroelectric crystals is conditioned by the physical nature of nonlinearity—it is associated with electronic polarization, and, accordingly, with the so-called electronic nonlinearity [14,15]. This mechanism is caused by the distortion of the outer electron cloud of atoms, ions, and molecules compared to their unperturbed state, and it has a very fast response time ($<10^{-15}$ s) but a low $\chi^{(2)}$. In [16], it is shown that the listed mechanism of nonlinearity can be enhanced by spatial redistribution of electrons due to a spatially modulated light flux, which is the characteristic of spatially resonant structures, such as photonic crystal fibers. However, the materials listed above have low values of $\chi^{(2)}$, which means that they will not allow one to obtain an efficient frequency transformation.

Regarding the materials with higher nonlinearity coefficients, one can mention organic materials [17], in which the value of $\chi^{(2)}$ can be in the range $\cong 10^{-6} \ldots 10^{-7}$ m/V, which is relatively large. However, the mechanism for achieving such values of $\chi^{(2)}$ refers to molecular reorientation with a very big establishment time about $\sim 10^{-10}$ up to $10^{-3}$ s, i.e., the greater the value of $\chi^{(2)}$ that can be reached, the longer one has to wait until the material enters the nonlinear regime, i.e., the longer the molecule, the greater the nonlinearity achieved, but also the longer this molecule turns. Therefore, the authors of studies in this

area focus not only on the values of the coefficients obtained but also on the dynamics of the observed processes. In addition, this class of materials exhibits a significant dependence of light transmission on its physical state and condition, for example, on humidity. There is also a significant oscillating dependence on frequency in the visible and near-IR radiation.

Nevertheless, in [17], it is said that these materials can be used to generate, for example, a difference frequency in the case when this is not associated with the signal modulation but is necessary only to obtain the indicated radio emission (in other words, when one wants to generate radio carrier frequency). The authors of [17] also show that $\chi^{(2)}$ depends significantly on the ratio of the molecular axis and the electric intensity vector of the input light field, and it can be found through relation Equation (5).

$$\chi_{kkk}^{(2)} \; = \; N f_{local} \beta_{kkk}^{eff} \; = \; N f_{local} \left\langle \cos^3 \theta_{kz} \right\rangle \beta_{zzz}, \tag{5}$$

where $\beta_{zzz}(-2\omega; \omega, \omega) \; = \; \frac{\omega_{eg}^4}{\left( \omega_{eg}^2 - 4\omega^2 \right)\left( \omega_{eg}^2 - \omega^2 \right)} \beta_0$ (see [17] for details).

From the point of view of the problem under consideration, in addition to the long time of the nonlinear regime establishment, a significant drawback of these materials is also the impossibility of their implementation in the form of extended waveguides in the form of fiber optic structures.

An interesting subclass of the considered category of materials is the so-called metal-organic frameworks (MOF) [18–21]. In MOFs, not only the organic filler is responsible for the macroscopic properties (see [17]), but also the metal structure, which, besides the structure strength (which was laid in the original idea of developing the MOF), can also form a field potential when irradiated with an electromagnetic field of the corresponding frequency range [18,22]. Strictly speaking, as we know, there are no results of experimental studies on the effect of the metal frame that comprises the MOF on the possible generation of plasmon field effects, which affects the macroscopic optical properties of the material. It should be noted that the nonlinear properties of MOF structures are studied quite intensively in both theoretical [23] and experimental [24] ways. For example, in [25], research is primarily focused on the design of a molecule that provides the required nonlinear properties.

According to publications [19,21], MOFs can be used to form an optical quadratic nonlinearity in problems of generating, for example, the second harmonic:

$$I_{2\omega} \; = \; \frac{32\pi^3 \omega^2 s^2 \theta_{2\omega}}{c^3} \left| e_{2\omega} \chi^{(2)} e_\omega^2 \right|^2 I_\omega^2 \infty \left( \chi^{(2)} \right)^2 I_\omega^2, \tag{6}$$

where the intensity of the second harmonic $I_{2\omega}$ is proportional to $(\chi^{(2)})^2$. The main properties of MOFs from the point of view of the problem under consideration include the following:

(1) Low dependence of light transmission on frequency in the visible and near-IR region [17], i.e., the transmission coefficient is almost constant in this frequency range;

(2) The $\chi^{(2)}$ is in the range $\cong 10^{-3} \dots 10^{-9}$ m/V, which is by several orders of magnitude higher than in ferroelectric crystals;

(3) Nonlinear regime establishment time is ~$10^{-14}$ up to $10^{-3}$ s (the spread is very significant because the time depends significantly on the influence of the metal frame);

(4) The properties of MOF can be controlled by an external field/potential;

(5) MOFs cannot be made in the form of extended guides/fiber optic structures.

In the framework of the considered problem, MOF under certain conditions can be used as the basis of a quadratic-nonlinear element. Namely, in the case of a decrease in the nonlinear regime establishing time (to values of $10^{-14}$–$10^{-13}$ s) due to the use, for example, of plasmon effects. There are many varieties of MOFs designed for both nonlinear optics and microwave radiation; the proper choice of MOF structure is a subject for further study; MOFs for nonlinear optics are described, for example, in [26].

Another option for constructing a quadratic-nonlinear element with properties suitable for the problem being solved is the use of a photonic-crystal fiber (PCF). It should be mentioned that PCFs have the widest range of macroscopic optical properties: in some cases, their nonlinearity is reduced to zero due to the propagation of the light field in areas filled with air [16]; in other cases, they obtain a significant macroscopic nonlinearity [16,27–31] achieved, for example, by channeling the light field in regions of ultra-small diameter, which provides a significant increase in the intensity and a corresponding increase in the effective nonlinear response. As an example, this option is used for spectrum broadening by the interaction of nonlinearity on short pulses (with durations of $\sim 10^{-12}$ s).

The results concerning the quadratic nonlinearity in PCF have been published widely [27–31]. So, in [29], it is said that $\chi^{(2)}$ can be in the range $\cong 10^{-3} \ldots 10^{-10}$ m/V, which many times exceeds the corresponding values for ferroelectric crystals. There are additional mechanisms for obtaining nonlinearity [27,28] associated with injected carriers, which are put into operation with a nonlinear shift in the photonic bandgap [27]. At the same time, this mechanism is inherently akin to electronic nonlinearity, which means that it is characterized by a short establishment time. Therefore, in quadratic-nonlinear PCFs constructed in the above manner, one can expect significant $\chi^{(2)}$ along with short entry times into the specified regime. As we know, there is no published data on the dynamic parameters of the nonlinear regime of PCF. It is also obvious that PCFs are waveguiding structures, which is undoubtedly of interest for the problem being solved.

It is mentioned in [30,31] that quadratically nonlinear PCFs can be successfully used to generate the second harmonic and difference frequency [32], including the THz range [33]. However, if a high nonlinearity is achieved by spreading the input light beam over high-intensity narrow light "filaments", then this will actually lead to multipole [34] generation of the radio frequency wave and, quite possibly, will distort the information being carried. Therefore, PCFs can potentially be used for the problem of square-nonlinear generation of the RF difference frequency, but the considerations presented above will require optimization of the PCF structure to reduce the distortion of the generated signal, as shown in the above-mentioned papers. The main properties of the considered nonlinear materials are summarized in Table 1.

**Table 1.** Comparison of the considered nonlinear materials.

| Nonlinear Material/Structure | $\chi^{(2)}$ Value, m/V | Nonlinear Response, s | Ability to Manufacture Long Guiding Structures, Focons and/or Structures with Refractive Index Gradient | Note |
|---|---|---|---|---|
| Ferroelectric crystals (PPLN, etc.) | $\cong (0.5 \ldots 45) \times 10^{-12}$ | $<10^{-15}$ | Waveguides are made in chip form | Well-known technology |
| Organic materials (e.g., DAST) | $\cong 10^{-6} \ldots 10^{-7}$ | $10^{-10} \ldots 10^{-3}$ | Possible | Widely used for THz generation |
| Metal-organic frameworks—MOF | $10^{-3} \ldots 10^{-9}$ | $10^{-14} \ldots 10^{-3}$ | Possible | Potential usage of plasmon effects |
| Photonic crystal fibers | $10^{-3} \ldots 10^{-10}$ | $\cong 10^{-15}$ | Difficult; it is necessary to avoid multipole wave addition | Usage of photonic bandgaps |

Thus, according to the presented discussion of materials and structures with quadratic-nonlinear optical properties, it can be concluded that MOFs and/or PCFs can be used for the problem being solved. In the first case, i.e., for MOF, one can either use the parameters of some known structure (material) to develop a non-linear element for DFG or study the effect of a metal frame and thereby optimize MOF parameters. In the second case, i.e., for PCFs, it is necessary to take into account the properties of the photonic-crystal structure that determine the mechanism of nonlinearity and, accordingly, the mechanism for generating a radio wave, and generally speaking, it is necessary to optimize the PCF's nonlinearity for the problem under consideration.

### 3. Proposed Conversion Device and Formulation of the Problem of Modeling Its Parameters

As can be seen from the discussion in Section 2, the simplest solution is the use of MOF, as presented in [21]. In this regard, to minimize losses and increase the efficiency of a radio-photonic conversion device designed to generate a radio frequency, it is necessary to construct a nonlinear element in the form of a guiding structure that retains resonant properties for both input optical and output radio radiation. This can be achieved by changing the diameter of said guided structure $D(z)$ along the propagation axis $z$ as in horn antenna or focon. In this case, for the input optical radiation $(\lambda_1, \lambda_2)$, it is desirable to maintain a few-mode (with the number of modes of the order of 5, which corresponds to low-order LP modes in weakly guiding fibers) propagation along the entire length $L$ of the nonlinear element. Or, in any case, to optimize the parameters of the nonlinear element so that the few-mode propagation regime for the input radiation is optimally maintained (i.e., there is no high-order mode excitation or no significant mode coupling). This can be achieved by changing the radial profile of the refractive index of the material along the length of the nonlinear element, i.e., providing the corresponding refractive index profile $n(z,r)$. Note that by "(quasi) single-mode" we mean the OAM mode (within the scalar approximation it corresponds to a set of linearly polarized (LP) modes containing angular exponent $\exp(i\ell\varphi)$, and each of these modes is a superposition of degenerated transverse modes [35]), so generally speaking, we are talking about few-mode regime. So, the proposed device, concerning the analog [2], is a focon—a guiding structure designed to guide radiation in a given direction. The use of a focon will make it possible to avoid a decrease in the radiation intensity when it is coupled from the fiber, in contrast to PPLN in the scheme, described in [2], which entailed a decrease in the efficiency of the nonlinear effect and, consequently, the transformation process. In addition, the proposed device is designed to provide a quasi-single-mode propagation mode inside the focon, due to a specially selected refractive index profile $n(r, z)$ and a function of changing the focon radius $f(r, z)$ from the coordinate $z$. This design will avoid dark areas because they are not involved in the conversion which means that this will also improve the efficiency of the proposed device.

Thus, the proposed radio-photonic conversion device should have the following structure (see Figure 2). Optical radiation is supplied to its input at two close wavelengths (e.g., 1550 nm and 1550.8 nm), the difference of which (~0.8 nm) corresponds to the generated radio wave of frequency about 100 GHz. Both input optical wavelengths should be fed through a few-mode fiber (FMF)—step-index fiber with relatively large core diameter about 16 µm. The connection between fiber and the focon can be provided by optical glue. The output part of this radio photonic device must have a substantially larger diameter than the input part, which is necessary to ensure effective resonant properties for the generated radio wave. Since the relationships that interconnect the geometric parameters of the guiding structure with the mode wavelength are linear, we can estimate:

$$\frac{D_{\text{out}}}{D_{\text{in}}} \approx \gamma \frac{\Lambda}{\lambda}, \tag{7}$$

where $\Lambda$ is the wavelength of the generated radio wave, $\lambda$ is one of the input optical wavelengths and $\gamma$ is a proportionality factor. Note that relation Equation (7) shows only the interdependence between the parameters but not the exact values. In order to obtain precise values, we provide a simulation of the focon taking into account its geometric and optical parameters (see Section 4).

Therefore, we can consider the nonlinear element as a focon, whose geometric profile $D(z)$ can either be a continuously increasing function or have a minimum (as shown in Figure 2) in the case of a significant inlet diameter. Such a focon can be made by MOF. The design of the focon parameters is reduced to find $D(z)$ and $n(z,r)$ with known $\lambda_1, \lambda_2, \chi^{(2)}$, $D_{\text{in}}, D_{\text{out}}$, and focon length $L$ provided that the amplitude of the output radio wave $A_\Omega$ is above a given threshold $A^*$: $A_\Omega \geq A^*$. An additional physical condition that ensures

$A_\Omega \geq A^*$, as mentioned above, is the requirement to maintain a quasi-single-mode regime for input radiation at length $H$, which in the limit $H \to L$. Note that if the focon is made of the PCF, the parameters of the PCF structure should also be included in initial design parameters, since the PCF's nonlinearity cannot be replaced by the averaged $\chi^{(2)}$. From a technological point of view, for the PCF-based focon manufacture, one can modify a photonic lantern technology (applied for PCF in, e.g., [36]) taking to account the nonlinear properties of the PCF; however, this problem requires a separate study and is beyond the scope of this article.

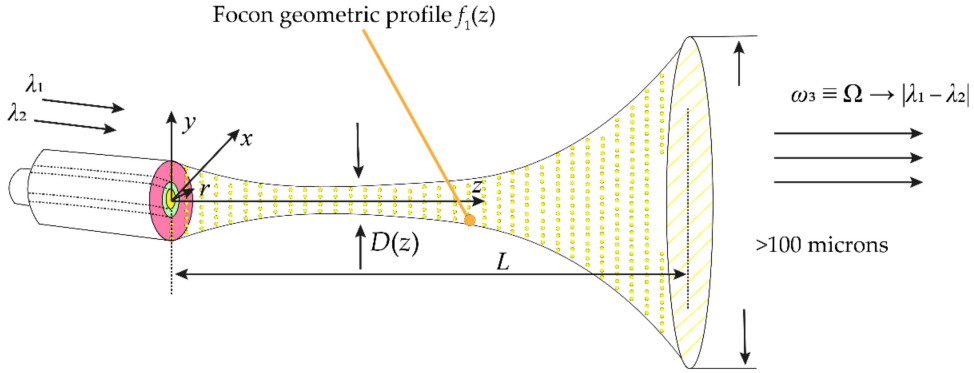

**Figure 2.** An illustration of the structure of the proposed radio-photonic conversion device that provides the generation of differential radio signal when optical radiation is supplied to the input at two close wavelengths.

Based on the proposed device and the design parameters, we need to model the electric field strength for the input (with indices 1 and 2) and output (3) waves to determine the spatial distribution of the refractive index of the focon and its geometric profile. We introduce the following parameters:

- For the focon profile, one can write $x^2 + y^2 = r^2 = f_1(z)$, and let $f_1(z)$ be a smooth function.
- $n(z, r)$ is the linear refractive index for $r^2 < f_1(z)$ inside the focon. In this case, we will assume that $n(r, z) = n_0 + \Delta n(r, z)$, and $\Delta n(r, z)$ determines the gradient of the refractive index, and in relative representation $\Delta n(r, z)$ is the same for each of the waves: $n(r, z) = n_0 + \theta(r, z)n_0 = n_0(1 + \theta(r, z))$, where $n_0$ is initial (unperturbed by the gradient) refractive index. Thus, the undetermined function for all waves is $\theta(r, z)$, and, as the results of similar studies presented in [37–40] show, $\theta(r, z)$ most likely will not exceed a few percent of $n_0$ (1–10%). For definiteness, we set: $|\theta(r, z)| \leq 0, 1 \cdot n_0\theta(r, z)$. Note that the function $\theta(r, z)$ must satisfy some obvious requirements, for example, it must be continuous and monotonic; such requirements are dictated by the condition of focon's physical realizability. In this way, $n(r, z) = n_0 + \Delta n(r, z) = n_0 + \theta(r, z)n_0$.
- The parameter $\chi^{(2)}$ can be considered homogeneous over the focon volume, since it changes slightly in the presence of a linear refractive index gradient. This assumption is due to the fact that the effect of the gradient of the linear refractive index on the phase of the signal during the interference process is much more significant than the effect of a change on the nonlinear process (see below). In other words, when the linear part of the refractive index changes, its nonlinear part undergoes proportional changes, which is, however, very insignificant. We also neglect the dispersion of $\chi^{(2)}$ for the considered waves according to [41].
- The refractive index outside the focon $n(r, z) = n_{cl}$ for $r^2 > f_1(z)$, where we assume that $n_{cl} = 1.46$ and index "cl" stands for "cladding".

## 4. Modeling the Parameters of the Proposed Device

The propagation of an electromagnetic wave in a nonlinear medium is described by the following equation:

$$\nabla^2 \vec{E} - \frac{\varepsilon}{c^2}\frac{\partial^2 \vec{E}}{\partial t^2} = \frac{1}{c^2}\frac{\partial^2 \vec{P}_{NL}}{\partial t}, \tag{8}$$

where $\vec{E}(x, y, z, t)$ is the vector of the electric field strength, $c$ is the speed of light in vacuum, $\varepsilon$ is the permittivity of the medium, and $\vec{P}_{NL}(x, y, z, t)$ is the nonlinear polarization vector.

By analogy with [2] and using the technique described in [41], to obtain a solution Equation (8), one should, firstly, use the separation of variables, and secondly, apply the Fourier transform of the given functions in the time domain. The latter allows us, in Equation (8), to get away from time derivatives by performing the following replacement, and thus, the $d/dt \rightarrow j\omega$ e Fourier image of the solution Equation (8) will be written in the form:

$$\widetilde{E}(x, y, z, \omega) = \widetilde{A}(z, \omega)\widetilde{F}(x, y)\exp(j\beta z) \equiv E(r, \varphi, z, \omega) = A(z, \omega)F(r, \varphi)\exp(j\beta z), \tag{9}$$

where we omit the symbol ~ in the further notation for simplification. Now we change the Cartesian system to cylindrical coordinates (we consider the focon to be a body of rotation); $\omega$ is the radiation frequency, $\beta = \frac{\omega}{c}\sqrt{\varepsilon}$, $A(z, \omega)$ is a function describing the longitudinal (along the focon, Figure 2 dependence of $E$, and $F(r, \varphi)$ is the transverse component of the vector $E$. In addition, in Equations (8) and (9), the indices 1, 2 and 3 related to the input (1 and 2) waves and the output Equation (3) wave, respectively, since the above expressions for all three waves appear to be identical.

Next, we find the derivative of Equation (9) with respect to coordinates, and substitute the resulting relation into Equation (8), taking into account the properties of the Fourier transform. Following [37], we neglect the second derivatives with respect to coordinates to obtain the final equation, since we assume that functions $A(z, \omega)$ and $F(r, \varphi)$ change much slower than $e^{j\beta z}$. Now, if we divide $\beta$ into components depending on the basic physical processes under consideration, we can obtain two equations of the form:

$$\frac{dA}{dz} = j[\beta_\Phi(\omega) - \beta_0] \cdot A \text{ and } \frac{dF^{(q)}}{dr} = j[\beta_\Phi(\omega) - \beta_0] \cdot F^{(q)}, \tag{10}$$

where $\beta_\Phi(\omega)$ determines the additional phase incursion of the wave under the influence of the determining physical process. So, for example, if a light wave propagates in an extended optical path, then the determining physical process will be chromatic dispersion. In this article, we assume that there are no dispersion distortions during the propagation of waves inside the focon since we consider two close wavelengths. On the contrary, quadratic-nonlinear changes in the refractive index have significant influence, characterized by $\chi^{(2)}$, when the waves propagate along the focon, and the interference of waves during transverse distribution, which ensures the addition of re-reflecting waves into a single mode. However, the function $F$ in (9) also has a $z$-axis dependence; moreover, as shown in [37–39], to obtain a proper solution, one should divide the entire focon into transverse layers of width $\Delta z$ where $\Delta z \rightarrow 0$ (see Figure 3). Within each layer, the "transverse" process can be considered independent of $z$, as noted in the second equation in (10) and justified in [37–39]. Here, index $q$ is the counter of such layers. The dependence of $F$ on $z$ is ensured by the fact that the output of each $q$th layer is the input of each $(q + 1)$th layer. In addition to previous assumptions, we also neglect the Rayleigh losses inside the focon since it has quite short length. The parameter $\beta_0$ in (10) determines the phase incursion of a wave in a medium with an initially unperturbed refractive index (in the absence of nonlinear processes and any refractive index gradient), i.e., $\beta_0 = \frac{\omega}{c}n_0$.

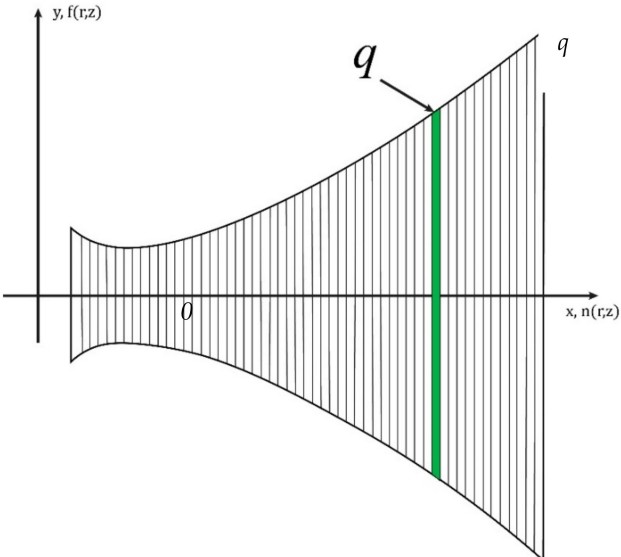

**Figure 3.** Illustration of the principle of splitting the focon into short layers, $q \in [1, Q]$, $Q$ is total number of layers, $z = \Delta z \cdot q$.

We make the following assumption: there is no re-reflection (and hence interference) between the layers in the longitudinal direction, despite the difference in the refractive index between the layers. This is because the focon has a smooth gradient and does not consist of layers with a hop in the refractive index at the boundary. Moreover, when $\Delta z \to 0$, the increment of the refractive index between the layers is $\Delta n \to 0$.

Further, if we take into account that in a quadratic-nonlinear medium $\vec{P}_{NL} = \chi^{(2)} E \cdot \vec{E}$, i.e., $P_{NL} = \chi^{(2)} |E|^2$, and make mathematical transformations similar to the technique [41] for the function $A$, and represent the function $F$ in the form of counter-propagating waves $F_\uparrow$ (up) and $F_\downarrow$ (down) relative to the focon axis, then for each of the waves (1, 2 and 3), Equation (10) will take a form that is suitable for numerical solution:

$$\frac{dA_1^{(q)}}{dz} = \varepsilon_0 \chi^{(2)} \frac{\omega_1^2}{2c^2} \left[ \left( \mu_{11}^{(q)} \left| A_1^{(q)} \right| + 2 \sum_{m \neq 1} \mu_{1m}^{(q)} \left| A_m^{(q)} \right| \right) A_1^{(q)} + 2\mu_{123}^{(q)} A_2^{(q)*} A_3^{(q)} \exp(j\Delta\beta z) \exp(j\Delta\ell\varphi) \right]$$

$$\frac{dA_2^{(q)}}{dz} = \varepsilon_0 \chi^{(2)} \frac{\omega_2^2}{2c^2} \left[ \left( \mu_{22}^{(q)} \left| A_2^{(q)} \right| + 2 \sum_{m \neq 2} \mu_{2m}^{(q)} \left| A_m^{(q)} \right| \right) A_2^{(q)} + 2\mu_{213}^{(q)} A_1^{(q)*} A_3^{(q)} \exp(j\Delta\beta z) \exp(j\Delta\ell\varphi) \right]$$

$$\frac{dA_3^{(q)}}{dz} = \varepsilon_0 \chi^{(2)} \frac{\Omega^2}{2c^2} \left[ \left( \mu_{33}^{(q)} \left| A_3^{(q)} \right| + 2 \sum_{m \neq 3} \mu_{3m}^{(q)} \left| A_m^{(q)} \right| \right) A_3^{(q)} + 2\mu_{312}^{(q)} A_1^{(q)} A_2^{(q)} \exp(-j\Delta\beta z) \exp(-j\Delta\ell\varphi) \right]$$

$$\frac{dF_{1,\uparrow}^{(l,q)}}{dr} = j\left(\beta_l^{(q)} - \beta_{1,r,z}\right) F_{1,\uparrow}^{(l,q)} + j\Re^{(q)}{}_1 \left| F_{1,\downarrow}^{(l,q)} \right| \text{ and}$$

$$\frac{dF_{1,\downarrow}^{(l,q)}}{dr} = j\left(\beta_l^{(q)} - \beta_{1,r,z}\right) F_{1,\downarrow}^{(l,q)} + j\Re^{(q)}{}_1 \left| F_{1,\uparrow}^{(l,q)} \right|$$

$$\frac{dF_{2,\uparrow}^{(l,q)}}{dr} = j\left(\beta_l^{(q)} - \beta_{2,r,z}\right) F_{2,\uparrow}^{(l,q)} + j\Re^{(q)}{}_2 \left| F_{2,\downarrow}^{(l,q)} \right| \text{ and}$$

$$\frac{dF_{2,\downarrow}^{(l,q)}}{dr} = j\left(\beta_l^{(q)} - \beta_{2,r,z}\right) F_{2,\downarrow}^{(l,q)} + j\Re^{(q)}{}_2 \left| F_{2,\uparrow}^{(l,q)} \right|$$

$$\frac{dF_{3,\uparrow}^{(l,q)}}{dr} = j\left(\beta_l^{(q)} - \beta_{3,r,z}\right) F_{3,\uparrow}^{(l,q)} + j\Re^{(q)}{}_3 \left| F_{3,\downarrow}^{(l,q)} \right| \text{ and}$$

$$\frac{dF_{3,\downarrow}^{(l,q)}}{dr} = j\left(\beta_l^{(q)} - \beta_{3,r,z}\right) F_{3,\downarrow}^{(l,q)} + j\Re^{(q)}{}_3 \left| F_{3,\uparrow}^{(l,q)} \right|$$

$$(11)$$

on each step $\Delta z$ with number $q$. In this case, due to the fact that the function $F$ was divided into two components (up and down), then solution Equation (9) should also take the corresponding form (for each of the waves):

$$E^{(q)}(r,z,\omega) = \left[ \sum_{l=1}^{K_l} \widetilde{F}_\uparrow^{(l,q)}(r) \exp\left(j\beta_{r,z}r\right) + \sum_{k=1}^{K_l} \widetilde{F}_\downarrow^{(l,q)}(r) \exp\left(-j\beta_{r,z}r\right) \right] A^{(q)}(z) \exp\left(-j\beta_{r,z}z\right). \tag{12}$$

In Equations (11) and (12), we use the following designations: $\mu_{11}$, $\mu_{22}$, $\mu_{33}$, $\mu_{12}$, $\mu_{13}$, $\mu_{21}$, $\mu_{23}$, $\mu_{13}$ and $\mu_{32}$ are the overlap integrals between the corresponding waves ($\mu_{iv}$ for each pair of $i$th and $v$th modes), while we assume that $\mu_{12} = \mu_{21}$, $\mu_{13} = \mu_{31}$ and $\mu_{23} = \mu_{32}$; $\mu_{123}$, $\mu_{213}$ and $\mu_{312}$—overlap integrals of all three waves: $\mu_{123} = \mu_{213} = \mu_{312}$; parameters $\Delta\beta = \beta_{0,3} + \beta_{0,2} - \beta_{0,1}$, where $\beta_{0,i}$, are determined for the corresponding frequencies $\omega_i$ and unperturbed values $n_{0,i}$ at these frequencies. The factor $\exp(j\Delta\ell\varphi)$ in Equation (11) is determined by the difference of the topological charges (an integer) $\ell_i$ of the vortex beams for each of the waves and represents the projection of the OAM in the direction of propagation [2–5]; it is determined by the dependence of the electromagnetic field of the vortex beam on the azimuthal angle $\varphi$. If the waves have no OAM, then the specified multiplier is not taken into account, i.e., $\exp(j\Delta\ell\varphi) = 1$ when $\Delta\ell = 0$. Otherwise, $\Delta\ell = \ell_1 - \ell_2 - \ell_3$ by analogy with [2]. The parameter $\beta_l^{(q)}$ is the wavenumber of the $q$th layer (see Figure 3), which has multipath interference properties. Each layer can be considered as a multibeam interferometer—a Fabry–Perot interferometer (FPI). However, due to the fact that the medium is nonlinear, $\beta^{(q)}$ depends on how many re-reflections have already occurred in this $q$th layer; here $l$ is the re-reflection counter, $l \in [1, K_l]$, $K_l$ in the ideal FPI is equal to $\infty$, but in this case it is determined by the number of effective re-reflections and is calculated through the reflection coefficient of the focon profile. For each of the waves, we will assume that:

$$\beta_l^{(q)} = \beta_M^{(q)} + \Delta B^{(q)}l, \ \beta_{MD}^{(q)} = \frac{\pi}{2\sqrt{f_1(\Delta zq)}} \langle n(r, \Delta zq)\rangle|_r,$$

$$\Delta B^{(q)} = \frac{2\pi\chi^{(2)}\left(\sqrt{P^{(q)}_1} + \sqrt{P^{(q)}_2} + \sqrt{P^{(q)}_3}\right)}{\lambda} \max(\mu_{i,m}^{(q)}), \tag{13}$$

where $\beta_{MD}^{(q)}$ is the initial (unperturbed) wave number of the $q$-th FPI with the averaged refractive index over the radius. This leads to the fact that the calculation of $F^{(l,q)}$ is performed for a certain FPI refractive index averaged over the radial gradient, and thus allows one to obtain only an estimations, but, on the other hand, ensures avoidance of the recurrence. Moreover, we apply this approach at the first iteration of the calculation; when the main parameters have already been estimated, in particular, an approximate value $\widehat{n}(r,z)$ has been obtained, then for the calculation of $\beta_{MD}^{(q)}$ in Equation (13) it is possible to use the value $\widehat{n}(r,z)$ found at the previous iteration. The parameter $\Delta B$ in Equation (13) determines the nonlinear contribution from the transmitted radiation, and is written by analogy with [38,39], $P^{(q)}_1$, $P^{(q)}_2$ and $P^{(q)}_3$ are the peak powers of the waves at the point $\Delta z \cdot q$, which at the first iteration can be calculated as $P^{(q)} = \left|E_0^{(q)}\right|^2$ for each of the waves. The parameter $\beta_{r,z}$ in (12) is defined as $\beta_{r,z} = \frac{\omega \cdot n(r, \Delta zq)}{c}$, but for the first iteration it is determined as $\beta_{r,z} = \frac{\omega}{c} \langle n(r, \Delta z \cdot q)\rangle|_r$. To calculate the coupling coefficient of interfering waves $\Re_i$, $i = 1, 2, 3$, similarly to [42], we will use the relation:

$$\Re^{(q)}_i \cong \frac{\pi}{\lambda_i} \left| \left( \langle n_i(r, \Delta z \cdot q)\rangle|_r \right)^2 - n_{0,i}^2 \right| \cdot \mu^{(q)}_{ii}, \tag{14}$$

according to which each of the waves interferes only with itself. In Equation (14), the index $i$ of the refractive index relates to the corresponding frequency (wavelength).

The number of effective re-reflections in the FPI can be determined as [42,43]: $K_l = \frac{\pi \cdot \sqrt{\rho_z}}{(1-\rho_z)}$ where $\rho^{(q)}$ is the reflection coefficient of the focon profile at arbitrary point $z$ (due to reflection from the interface between media with different refractive indices): $\rho^{(q)} = \frac{\left| n\left(r = \sqrt{f_1(\Delta z \cdot q)}, z\right) - n_{cl} \right|}{n\left(r = \sqrt{f_1(\Delta z \cdot q)}, z\right) + n_{cl}}$. The overlap integrals $\mu$ for the considered waves (for each pair of $i$-th and $v$-th modes) are determined by analogy with [38]:

$$\mu_{i,v}^{(q)} = \frac{\left\langle \left( F_{\uparrow,i}^{(q)} + F_{\downarrow,i}^{(q)} \right) \left( F_{\uparrow,v}^{*(q)} + F_{\downarrow,v}^{*(q)} \right) \right\rangle}{\sqrt{\prod_i \left\langle \left| F_{\uparrow,i}^{(q)} + F_{\downarrow,i}^{(q)} \right|^2 \right\rangle}}, \tag{15}$$

where the terms $F_{\uparrow,i}^{(q)}$ and $F_{\downarrow,i}^{(q)}$ for each of the waves can be found using the properties of the FPI for re-reflecting (superposing) waves [40]:

$$F_\uparrow^{(q)} = \sum_{l=1}^{K_l+1} \left( \rho^{(q)} \right)^{l-1} (\Delta z \cdot q) F_\uparrow^{(0,q)} \exp\left( j(l-1)\delta^{(q)} \right);$$

$$F_\downarrow^{(q)} = \sum_{l=1}^{K_l+1} \left( \rho^{(q)} \right)^{l-1} (\Delta z \cdot q) F_\downarrow^{(0,q)} \exp\left( j(l-1)\delta^{(q)} \right); \quad F^{(q)} = F_\uparrow^{(q)} + F_\downarrow^{(q)}. \tag{16}$$

The parameter $\delta$ will be defined by analogy with (13): $\delta^{(q)} = \frac{4\pi \cdot \sqrt{f_1(\Delta z \cdot q)}}{\lambda} \cdot \langle n(r, \Delta z \cdot q) \rangle|_r$, and the initial amplitude value will be set for the first iteration as: $F_\uparrow^{(0,q)} = F_\downarrow^{(0,q)} \cong \frac{1}{4}\sqrt{P^{(q)}}$, assuming that the refraction index is isotropic within the layer and therefore the power is divided equally, i.e., a half for each wave. In addition, for the components of the terms $F$ under consideration, which characterize re-reflections from the focon profile in the transverse direction, the following one can use the following relations:

$$\frac{F_\downarrow^{(l,q)}}{F_\uparrow^{(l+1,q)}} = \rho^{(q)} = \frac{F_\uparrow^{(l,q)}}{F_\downarrow^{(l+1,q)}}, \quad F_\downarrow^{(K_l,q)} = F_\downarrow^{(0,q+1)}, \quad F_\uparrow^{(K_l,q)} = F_\uparrow^{(0,q+1)}. \tag{17}$$

The transformation in Equations (8)–(11) is generally known, so we considered it unnecessary to present a detailed derivation of equations Equation (11) in the article. Let us pay attention to the fact that the left parts of each equation for the functions $A$ in Equation (11) do not depend on the angle $\varphi$, so the solutions of system Equation (11) exist under the following condition: $\Delta\ell = 0$. This suggests that in a quadratic-nonlinear medium, the input signals carrying OAMs $\ell_1$ and $\ell_2$ at frequencies $\omega_1$ and $\omega_2$ generate an output signal with OAM $\ell_3 = \ell_1 - \ell_2$ and frequency $\omega_3 = \omega_1 - \omega_2$, which ensures the seamlessness of optical-radio engineering conversion. Note that for the existence of OAM modes, as is well known [35], a few-mode regime (excitation of several transverse modes) is required. That is why we search $n(r, z)$ and $f_1(z)$—to provide a proper few-mode regime. Therefore, we reckon that the wave field (for each of the waves under consideration), determined by Equation (12), should obey the following relation:

$$E_{M-1}^{(q)} = \sqrt{P^{(q)}} \frac{J_\ell\left( u_{\ell m} \frac{r}{\sqrt{f_1(\Delta z \cdot q)}} \right)}{J_\ell(u_{\ell m})} \cdot e^{-j\beta^* z} e^{-j\omega t} e^{\ell\varphi}, \tag{18}$$

where $J_\ell$ is the Bessel function of the $\ell$-kind, $u_{\ell m}$ is the first $\ell m$th maximum of the $J_\ell$, $P^{(q)}$ is the power parameter, and $\beta^*$ is the generalized mode propagation coefficient. The specified "quasi-single-mode" addition will ensure the absence of dark areas, which means the large volume of the interacting light, and with this the highest efficiency of the device (especially

in case of low-order OAMs, e.g., $\ell_1 = 1$, $\ell_2 = 0$, etc.). Relation Equation (18) is valid for $E$ as a function of time, to which inverse Fourier transform Equation (12) should be performed.

Note that the stated condition can be simplified, namely, the power parameter in (18) can be considered only as a multiplier, i.e., as weight coefficient. In addition, the considered amplitude distribution has no dependence on $z$ (except the phase exponent), but only on $r, \varphi$, so condition Equation (18) applies only to function $F$. Therefore, we consider that relation Equation (18), which is valid for $F^{(q)}$ as a function of time, complements relations (16). As a result of the numerical solution of equations Equation (11), taking into account Equations (13)–(18), iterative selection of the functions $n(r, z)$ and $f_1(z)$ was carried out (see Figure 4). In the calculations, we assumed $\chi^{(2)} = 10^{-6}$ m/V. As we can see in Figure 4, the proposed converter structure can be easily coupled with few-mode fiber due to a comparative input diameter of a few microns.

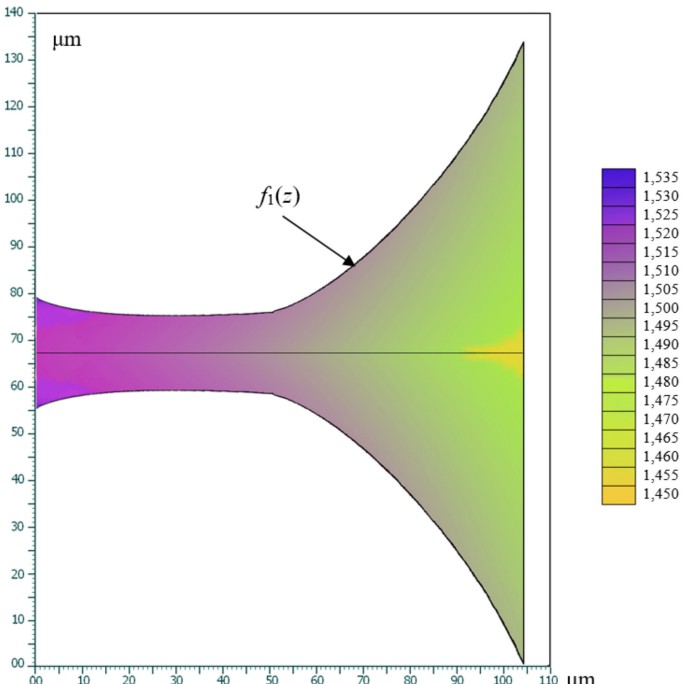

**Figure 4.** An illustration of the calculated parameters (distribution of the refractive index gradient defined by $f_1(z)$) at the second iteration of the calculation under the condition that the power of the generated wave $P_3$ is at least 1% of $P_1$ (see an explanation below).

The purpose of the simulation was to find the parameters $n(r, z)$ and $f_1(z)$; we have carried out two iterations. For the first iteration, we have set all the input values described above (see explanations to Figure 2) as well as the initial values of the transverse amplitude components: $F_\uparrow^{(0,q)} = F_\downarrow^{(0,q)} \cong \frac{1}{4}\sqrt{P^{(q)}}$, where $P^{(q)}$ for $q = 1$ was taken equal to 30 mW, which is an average fiber optics signal power (we assume the fiber optic input of the device). The first iteration was intended to estimate a number of parameters, including the value of $P^{(q)}$. To ensure the acceptable level of output radio signal, the input signal powers were set as $P_1 = P_2 = 70$ mW at wavelengths $\lambda_1$ and $\lambda_2$, respectively. We assume that the desired parameters $n(r, z)$ and $f_1(z)$ must provide the following condition: $P_3 \geq 1\% P_1$, i.e., the output radio signal power $P_3$ is at least 1% of $P_1$ (see Figure 4).

The result of the first iteration proved the convergence of the model, and it was also found that $F_\uparrow^{(0,1)} = F_\downarrow^{(0,1)} \cong 1.32$ [mW]$^{1/2}$. After that, the second iteration was carried out to obtain the parameters $n(r, z)$ and $f_1(z)$. Figure 4 shows the obtained results.

The calculation showed that the condition $P_3 \geq 1\% P_1$ can be provided if the refractive index $n(r, z)$ in the $z$-axis of the entire focon changes very significantly—from 1450 to 1530. However, it basically changes in a volume constituting about 95% of the total volume of the

focon—by a value from 1475 to 1520. The function $f_1(z)$ provides narrowing of the focon region in its input (at a length of approximately 50 µm), and then this function undergoes a characteristic inflection, after which the focon noticeably expands. This narrowing is most likely explained by the need to first increase the intensity of the wave in order to effectively implement the nonlinear optical effect for the generation of $P_3$. Further, when the power of the generated wave reaches a sufficient value, the need for further concentration of the radiation density is already reduced. Another process becomes important—it is required to guide (channel) the generated wave along the *z*-axis, preventing its leakage through the sides of the focon.

It can be seen that the optical geometry of the focon (optical lengths equal to the product of geometric lengths and the value of the refractive index near them) tends to some leveling of the focon geometry, i.e., where the focon narrows, the refractive index is on average larger, and vice versa. It can be said that the focon turned out to be approximately similar to the guiding structure, i.e., waveguide. However, for nonlinear calculations, the presence of a narrowing region is essential, which, as noted earlier, increases the intensity (as well as the power flux) of the waves there, and hence it enhances the efficiency of their nonlinear interaction. This is the essential difference between a focon made of a nonlinear optical material and the traditionally used geometry of a PPLN crystal in similar tasks.

Undoubtedly, the parameters of the resulting focon, first of all—$n(r, z)$—are difficult to implement in practice. We believe that this issue is the subject of our further research; our goal here is to show the conceptual possibility of nonlinear focon converter. Nevertheless, a significant advantage of the calculated element is the possibility of obtaining an increased power of the generated difference wave in comparison with other solutions: 1% versus 0.1% using PPLN [11]. In addition, the dimensions of the proposed focon are significantly smaller than the dimensions of a typical PPLN crystal in similar problems: fractions of a millimeter versus several millimeters (ten millimeters). This potential focon subsequently can be integrated into a single fiber optic path, or into an integrated optical device, which will ensure the miniaturization of such a nonlinear optical-radio converter along with its high efficiency. Such a converter seems relevant for the seamless conversion of complexly shaped optical signals into the radio range, such as beams with OAM. Note that beams with OAM can form a group signal, i.e., the result of adding various OAMs, the decomposition of which before conversion would be very problematic (the need to use free space elements, complex adjustment, etc., which makes the practical use of such a system almost impossible). The use of the proposed focon will make it possible to avoid these difficulties, i.e., will provide fiber-optic-radio-air systems with OAM wide practical applications.

If we compare the proposed MOF-based focon with a similar device that can be made from a photonic crystal structure, it should be noted that the latter option can hardly be used to convert complexly shaped optical signals, in which the spatial phase of optical radiation is an important parameter. This is due to the fact that the high nonlinearity of photonic-crystal structures is achieved by using several thin optical-conducting paths that provide high radiation intensity, and with this, a significant nonlinear interaction of waves. At the output of such a nonlinear photonic-crystal device, optical beams (after the nonlinear processes) merge into a single output beam, where the required power results are achieved. However, from the point of view of the formation of a given output phase (leading to the specific distribution of the phase field, typical for OAM), significant distortions are likely to occur because of need of the phase matching. This allows us to conclude that MOF in the problem under consideration is preferable to a nonlinear photonic crystal structure. The simulation results obtained here are, in fact, the requirements for the MOF structure, on the basis of which a specific material should be selected.

## 5. Conclusions

In this article, we provide an analysis of currently known materials and structures with quadratic-nonlinear optical properties. These materials and structures can be used in the

process of non-linear parametric generation of the difference frequency in a two-frequency optical-radio engineering conversion.

The principle of constructing a nonlinear optical-radio-technical converter based on optical focon is proposed. Using the assumption that this focon can be made by MOF, a technique for modeling its parameters is proposed. The mathematical model of the process of propagation and nonlinear interaction of waves inside the focon is based on a simplification of the nonlinear wave equation.

Within the framework of the developed model, the following parameters are approximately determined: the 3D gradient of the linear refractive index and the function characterizing the geometric profile of the focon, which provide quasi-single-mode generation of the difference frequency. The proposed scheme can be used when deploying fiber-optic communication lines (FOCL) network segments with seamless optical-radio conversion (including vortex-signal conversion), which will greatly increase the capacity and throughput of telecommunication systems.

**Author Contributions:** Conceptualization, formal analysis and methodology, I.L.V. and A.R.G.; funding acquisition, formatting, I.K.M. and A.V.B.; software and visualization, I.L.V.; writing—original draft preparation, review and editing, I.L.V., A.R.G., I.K.M., A.V.B. and M.T. All authors have read and agreed to the published version of the manuscript.

**Funding:** This work was partially supported by the Ministry of Science and Higher Education of the Russian Federation for research under the State Assignment of FSBEI HE USATU No. FEUE-2020-0007 on the topic "Theoretical foundations of modeling and semantic analysis of the processes of transformation of vortex electromagnetic fields in infocommunication systems" in the part "Analysis of optical quadratic-nonlinear materials and structures potentially suitable for the design of a nonlinear converter" and partially funded by RFBR, DST, NSFC and NRF according to the research project 19-57-80016 BRICS_t in the part "Modeling the parameters of the proposed device".

**Institutional Review Board Statement:** Not applicable.

**Informed Consent Statement:** Not applicable.

**Conflicts of Interest:** The authors declare no conflict of interest.

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
