# Peer review of "A Nonlinear Radio-Photon Conversion Device"

_photonics, doi:10.3390/photonics9060417_

Round 1

Reviewer 1 Report

Phase matching process which is needed to obtain high conversion efficiency by using three-wave generation is not discussed in the paper. Could you please comment it and give potential solutions to implement phase matching?

The dispersion seems not to be taken into account in the theoretical approach. Could you include this parameter and comment on the benefit of this integration?

Author Response

Dear reviewer, all the corrections are highlighted by color yellow

Reviewer 2 Report

Authors need to explore more related works in the introduction section.

Why some sentences are highlighted with yellow color?

Authors need to present a comparison table to prepresent the improvement of this work.

Overall presentaion of this manuscript is excellent and can be accepetd after minor correction.

Author Response

Dear reviewer, here we present a response to your remarks:

Authors need to explore more related works in the introduction section.

Authors’ response: we have added some works describing photomixers

Why some sentences are highlighted with yellow color?

Authors’ response: since our manuscript is a resubmission, we’ve highlighted with color yellow parts corrected according to editor’s remarks

Authors need to present a comparison table to prepresent the improvement of this work.

Authors’ response: dear colleague, all the improvements are actually highlighted with color yellow

Reviewer 3 Report

In the manuscript under review, the authors proposed the design of a nonlinear optical converter in the form of a focusing cone, made of a metal-organic framework, for generating a difference frequency in the terahertz and sub-terahertz ranges. The authors have developed a method for calculating the optical and geometrical parameters of such a focon affixed at the output of an optical fiber used to deliver biharmonic laser pumping. Calculations show the possibility of generating a difference frequency with an efficiency of more than 1% of the power of one of the components of the biharmonic pumping.

I have read this article with great interest. In general, the work was done at a high scientific level. Of particular value to the readers is the second part of the article, devoted to the analysis of optical quadratic-nonlinear materials and structures potentially suitable for designing a nonlinear converter. However, in my opinion, the part of this material set out at lines 99-119, including Figure 1, can be replaced without compromising understanding by one or two sentences summarizing the experiment of the authors. Of course, Figure 1 is beautifully done, but it does not add anything aiding the reader in understanding the essence of the problem under consideration.

The authors themselves admit that the parameters of the focon calculated by them are difficult to achieve in practice. However, this does not diminish the scientific value of the work performed.

As a wish, I would like to suggest that the author slightly modernize the annotation indicating the efficiency of the nonlinear optical converter calculated by them.

There are some typographical errors in the text of the article, for example, at line 392 the nonlinear polarization in vector form should be written appropriately; at line 485 the meaning of the separating comma is not clear: “1,325 (mW)-1/2”. It is also not clear why the value of this number is given with an accuracy of up to the third decimal place after the decimal point. In some formulas there is an optional multiplication sign “⋅”, but in many expressions this sign is not present.

Author Response

Dear colleague, thank you for your review of our manuscript. We have finalized the article in accordance with your remarks, in particular, the annotation has been changed and errors have been corrected. Note that we added the first figure to illustrate the methodology of the experiment and show the reader live photos of the experimental setup; in our opinion, this should clarify the results described in the article. All the corrections are highlighted by color yellow.

Author Response

(The authors gave the same response as above.)

Round 2

Reviewer 1 Report

I agree to accept the publication of this paper

Reviewer 4 Report

Thanks to the authors for their collaboration in this updated version.

for the most of comments.

1. Still, the reply to comment 4 (corresponding to comment number 6 in the reviewer report) is not clear. The reply means that the expression 7 is just an approximation and the accurate value is obtained through the simulation as in Fig. 4. However, the difference between the two results is around 100 times so in this case expression 7 can not be claimed as an approximation or at least there is a missing factor in the expression. Also, the authors should clarify this point in the manuscript too not only in the review letter because anyone who wants to replicate the work would follow this expression to estimate the initial focon dimensions and would not expect that the optimized value would be much far away from the estimated one.

2. I think the authors may have mistakenly overlooked some of the comments from the previous round (only reply of 5 comments out of the 7). And still, the misleading conclusion that LiNbO3 crystal can only produce an extremely-low frequency conversion efficiency (η) in the order of 10-16 exists in the manuscript despite that using the same crystal (LiNbO3) and the same difference frequency generation (DFG) technique, Whispering Gallery Mode (WGM) LiNbO3 can experimentally produce a much higher efficiency in order of 10-5.

Could the authors comment on such a big difference in the efficiency or make it clear for the readers.

Author Response

Dear colleague, please check our response in the attached file
